# VUT: Versatile UI Transformer for Multi-Modal Multi-Task User Interface Modeling

## Abstract

User interface modeling is inherently multimodal, which involves several distinct types of data: images, structures and language. The tasks are also diverse, including object detection, language generation and grounding. In this paper, we present VUT, a Versatile UI Transformer that takes multimodal input and simultaneously accomplishes 5 distinct tasks with the same model. Our model consists of a multimodal Transformer encoder that jointly encodes UI images and structures, and performs UI object detection when the UI structures are absent in the input. Our model also consists of an auto-regressive Transformer model that encodes the language input and decodes output, for both question-answering and command grounding with respect to the UI. Our experiments show that for most of the tasks, when trained jointly for multi-tasks, VUT substantially reduces the number of models and footprints needed for performing multiple tasks, while achieving accuracy exceeding or on par with baseline models trained for each individual task.

## 1 Introduction

Modern graphical user interfaces specifically touchscreen mobile UIs enable a rich problem space for modeling where the input is inherently multimodal, which consists of several distinct types of data. A user interface screen exists in both a visual form, i.e., a screenshot, and a structural representation, i.e., a tree-like view hierarchy. Based on graphical user interfaces, there is a wide spectrum of modeling tasks that either directly enhance user experiences or advance the development of intelligent user interfaces. For example, previous work developed models and datasets for grounding a language command into an executable UI action (Li et al., 2020a), generating language description for accessibility on mobile devices (Li et al., 2020b; Wang et al., 2021), and understanding the usability of user interfaces (Swearngin & Li, 2019) or identifying the objects on the screen (Zhang et al., 2021). Previous work has also started learning effective representation of user interface screens (He et al., 2020; Li et al., 2021a), which can potentially benefit downstream tasks.

Although these previous works have made progress in addressing individual problems, it is important to investigate the feasibility of learning all these tasks with a single model. In addition to achieving a scientific understanding of how these UI tasks are related, it is extremely valuable to obtain such a multi-task model, which can potentially reduce the number of models that need to be developed and deployed. This is crucial for mobile devices that have limited computing resources. In this work, we propose VUT—Versatile UI Transformer, which handles three types of data: images, structures (view hierarchies) and language, and simultaneously performs five unique tasks that are representative in the UI modeling literature, including UI object detection, natural language command grounding, widget captioning, screen summarization and UI tappability prediction.

A major challenge we need to address is how to unify these distinct tasks as well as their heterogeneous datasets such that they can be learned by a single model. To this end, we devise a general formulation for UI modeling tasks based on five inherent types of information that define a task. We also aim to design a compact model architecture such that it remains stable for addressing a diverse and potentially growing set of tasks, for which we make each model component multi-purpose. Specifically, VUT comprises two Transformer architectures (Figure 1): the *Image-Structure* model, and the *Question-Answer* model. The Image-Structure model encodes the entire screenshot of a UI along its view hierarchy tree, with early fusion of the two modalities, which is guided by a *focus map* when a given object is inquired. In addition to being the UI encoder, the Image-Structure model

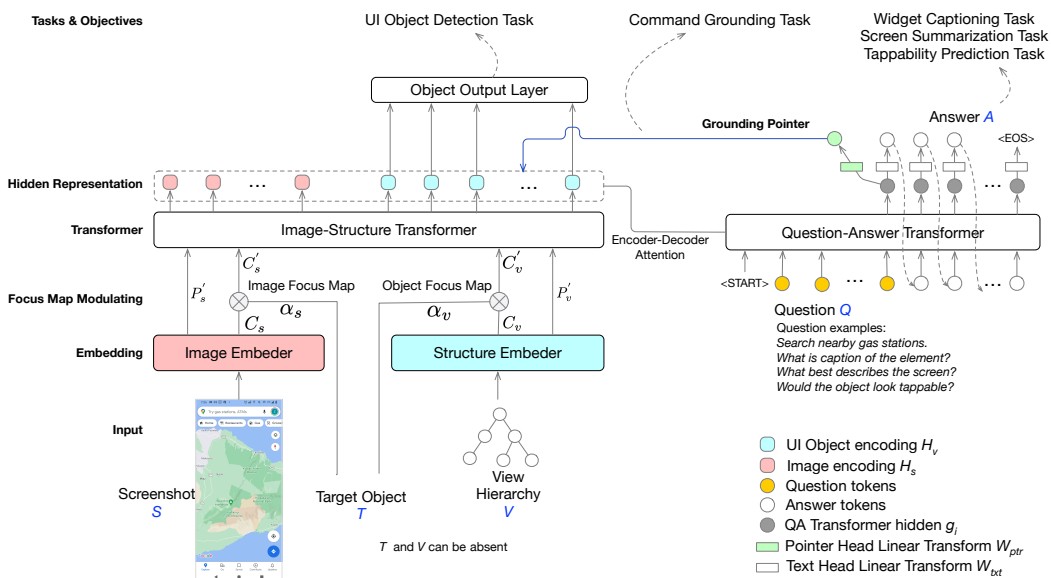

Figure 1: The VUT model architecture contains two Transformer models, which take image, structure and language input, and three task heads for achieving five distinct UI modeling tasks.

predicts UI objects when the view hierarchy is absent on the input,which achieves the UI object detection task. The Question-Answer model encodes a question while attending to the UI encodings from the Image-Structure model. It decodes a text answer when the task response is language, e.g., widget captioning (Li et al., 2020c) or screen summarization (Wang et al., 2021). For the grounding task whose output is an object reference, the Question-Answer model serves as the question encoder and its hidden state is used to locate UI elements to be acted upon. We highlight the relation of VUT with previous works in Table 1, and discuss their differences further in the following section.

We experiment with our model on 5 datasets, and compare the accuracy of VUT when it is trained alone for each task and jointly with multiple tasks. Our experiments show that VUT is able to perform all the five tasks simultaneously and achieve the performance on par with or surpass that when each task is learned alone. The main contributions of our work are as follows.

- We formulate multi-modal multi-task learning for a new domain—graphical user interfaces— with one model to accomplish a wide range of tasks for enhancing mobile user experiences.

- We design VUT based on a two-tower Transformer architecture, one for handling image-structure and the other for language data, where each Transformer is multi-purpose by both encoding and decoding its own modality, with cross-tower attention.

- We experiment with VUT on 5 distinct UI tasks, and thoroughly investigated the effect of these tasks when learned alone or jointly with ablations and analysis, which show the feasibility for achieving diverse UI tasks using a single model, which offers the value for reducing the number of models and storage footprints needed for deployment (Appendix E).

## 2 RELATED WORK

Extensive work has been conducted in multi-modal modeling with vision and languages (Li et al., 2019; Lu et al., 2019; Hu & Singh, 2021; Lu et al., 2020; Tan & Bansal, 2019; Kim et al., 2021; Zhou et al., 2020; Gupta et al., 2021). Existing works differ in the form of input they consume and the objectives of modeling. One category of work focuses on pretraining to learn an effective cross-modality representation for downstream tasks and the other directly learns multiple tasks end-to-end (Table 1). VUT belongs to the latter. In terms of the forms of multimodal data these models consume, most models handle image and text input. However, a unique form of data in UI modeling is the structure input of view hierarchies, which only VUT, UIBert and ActionBert use.

| Model | Image input | Structure input | Text input | Objectives |
|---|---|---|---|---|
| ViLBERT (Lu et al., 2019) | Object regions | None | Object captions | Pretraining |
| UIBert (Bai et al., 2021) | Object regions | View hierarchy | Object text | Pretraining |
| ActionBert (He et al., 2020) | Object regions | View hierarchy | Object text | Pretraining |
| ViLT (Kim et al., 2021) | Entire image | None | Image caption | Pretraining |
| VLP (Zhou et al., 2020) | Object regions | None | Image caption | Pretraining |
| LXMERT (Tan & Bansal, 2019) | Object regions | None | Image caption | Pretraining |
| 12-in-1 (Lu et al., 2020) | Object regions | None | Task prompts | Multi-task |
| UniT (Hu & Singh, 2021) | Entire image | None | Task prompts | Multi-task |
| GPV-I (Gupta et al., 2021) | Entire image | None | Task prompts | Multi-task |
| VUT (our model) | Entire image | View hierarchy | Task prompts | Multi-task |

Table 1: Comparison of VUT with several existing multi-modal modeling works.

Many existing works feed object regions, instead of the entire image to the model, which requires a pretrained object detection model (Lu et al., 2019; Bai et al., 2021; He et al., 2020; Zhou et al., 2020; Tan & Bansal, 2019; Lu et al., 2020) or address tasks only regarding the entire image (Kim et al., 2021; Hu & Singh, 2021). Although ActionBert (He et al., 2020) and UIBert (Bai et al., 2021) also address the UI domain, they are targeted for representation learning, and do not support multiple tasks simultaneously. As a result, they do not deal with language input of task descriptions. Their text input is those scraped from the UI screen, e.g., using OCR. In addition, these models require predetermined object regions similar to many BERT-based multi-modal models. In contrast, object detection is one of the tasks that VUT addresses.

In terms of modeling techniques, we designed a novel Transformer architecture for multi-task modeling of the UI domain, based on building blocks previously proposed for natural images and language, e.g., (Hu et al., 2020; Lu et al., 2020). the work that is closely related to ours is GPV-I (Gupta et al., 2021), which uses DETR (Carion et al., 2020) for object detection, and ViBERT (Lu et al., 2019) for multimodal modeling. In addition to the obvious deviation our work, e.g., VUT uses structure input but GPV-I does not, there are several important architecture differences. While GPV-I directly embeds DETR, an encoder-decoder model, into its architecture, VUT uses a single tower design where both the image and object queries are fed to the same Transformer encoder. This design choice is motivated by our goal to achieve a compact architecture, which the Image-Structure model serves both image-structure encoding and object detection (when the structure input is absent in the input). As shown in our experiment, the single tower architecture of VUT's Image-Structure model showed clear advantage over the encoder-decoder architecture in DETR for the UI object detection task. To address the unique domin of UI tasks, we also introduce focus map to guide the model towards the object being inquired. VUT's question-answer Transformer is designed based on existing auto-regressive multi-task language models (Raffel et al., 2019; Brown et al., 2020) where a question or a command is fed to the model as a prefix, and the responses are decoded token by token. One difference is that for the language command grounding task, instead of generating a language response, the last hidden state of the language model is used, as a question encoding, to retrieve a UI object on the screen.

## 3 PROBLEM FORMULATION

A graphical user interface contains a collection of UI elements for fulfilling a coherent set of tasks. There are often five types of data involved to formulate a UI task: $< S, V, T, Q, A >$ (Figure 1). $S$ is the screenshot image that captures the visual appearance of the UI screen. $V$ is the view hierarchy tree that represents the underlying structure of the screen. $T$ is the target object on the screen to be inquired. $Q$ is the natural language description of the task, which can be an open-ended question such as "*What is the caption of the element?*", a yes-or-no question such as "*Does the object look clickable?*" or a command such as "*Click on the Next button.*". See the full list of $Q$ used in our experiments in Appendix A. Finally, $A$ is the natural language answer to the question $Q$ when the form of the response for the task is supposed to be natural language. Depending on each task setup, these data types appear as either input or output. We elaborate on the formation of each task here, and use $\mathcal{F}$ to denote the function for achieving each task.

### 3.1 UI OBJECT DETECTION

Given the screenshot image, $S$, the task is to detect each UI element on the screen, such as Text Field, Toggle Button, or Image View. This task is similar to the typical object detection task in natural images (Carion et al., 2020) or recent UI object detection work (Zhang et al., 2021). However, our task is more challenging in that it needs to detect different types of container objects, which determine how UI objects are visually structured of the screen, such as Linear Layout, Frame Layout or List. In total there are 21 types of leaf or non-leaf objects in a view hierarchy. See Appendix D for the full list of objects we detect. UI object detection is important for improving accessibility and enabling other intelligent features such as UI adaptation when view hierarchy is not available. As a screen understanding task, UI object detection is beneficial to other UI modeling tasks as we will show in our experiments. The task is formulated as the follow (Equation 1).

$$V = \mathcal{F}(S, V_\varnothing, T_\varnothing, Q_\varnothing) \tag{1}$$

Note that this task is achieved solely based on the single-tower Image-Structure Transformer (Figure 1) and does not rely on the question-answer model. $V_\varnothing$, $T_\varnothing$ and $Q_\varnothing$ represent each type of data masked out or missing in the input.

### 3.2 WIDGET CAPTIONING

Generating natural language description for user interface elements is important for accessibility [1] and language-based interaction in general. The widget captioning task was initially proposed by Li et al. (2020b) and it extends the classic image captioning tasks (Xu et al., 2015) to the UI domain. In this task, given the UI view hierarchy, $V$, the screenshot image, $S$, and the target element to be captioned, $T$, the model predicts a natural language phrase, $A$, that best describes the functionality of the object (Equation 2).

$$A = \mathcal{F}(S, V, T, Q) \tag{2}$$

The model uses the information of $S$, $V$ and $T$ via the Image-Structure model. The examples of $Q$ are "What is the caption of the element?" and "What best describes the object?", and the examples of $A$ are "Forward", and "Shopping Cart".

### 3.3 SCREEN SUMMARIZATION

Instead of focusing on an individual element as the widget captioning task. screen summarization that is recently proposed by Wang et al. (2021) is the task that describes the entire UI screen (Equation 3) by producing a summary phrase.

$$A = \mathcal{F}(S, V, T_\varnothing, Q) \tag{3}$$

The examples of task prompts, $Q$, for screen summarization are "What is the description of the screen?" and "What best summarizes the UI?" $Q$ signals the model the type of responses it should generate when it is jointly trained with other tasks such as widget captioning. The screen summarization task is broadly related to multimodal summarization tasks in the literature, but is specific to the user interface domain.

### 3.4 LANGUAGE COMMAND GROUNDING

An important feature of modern smartphone interfaces is to interpret the natural language command of users as executable actions, e.g., Voice Control [2]. Previous work has investigated language grounding on user interfaces (Pasupat et al., 2018; Li et al., 2020a). In this task, given the UI, $S$ and $V$, and the language command, $Q$, the model needs to predict which object on the screen can fulfill the command, which is the opposite to the two preceding tasks of language generation.

$$T = \mathcal{F}(S, V, T_\varnothing, Q) \tag{4}$$

---

[1] https://support.google.com/accessibility/android/answer/6283677?hl=en
[2] https://support.apple.com/en-us/HT210417

Note that instead of generating a natural language response like widget caption and screen summarization, this task locates the target object, $T$, on the screen. In our dataset, we asked labelers to refer to an object on the screen in an unconstrained manner. The possibility of $Q$ is unbounded, which can be any phrase the user feels like using for manipulating the UI, e.g., "Go to the next screen", or "Tap on the checkout button". A command can also refer to an object indirectly using the relation to others, such as "Click the icon to the right of the search box."

### 3.5 Tappability Prediction

Lastly, we include a task related to automatic usability assessment. Whether a user perceives a UI object as tappable is an important usability issue. The mismatch between the tappability of an object as perceived by the user and its actual clickability has constantly plagued mobile user experiences. Swearngin & Li (2019) proposed tappability prediction as a binary classification task based on the view hierarchy and the screen appearance of the element. In this work, to avoid introducing additional task heads, we formulate tappability prediction as a yes-or-no QA task (Equation 5).

$$A = \mathcal{F}(S, V, T, Q) \tag{5}$$

which outputs "yes" when the perception of object is predicted as tappable, and "no" otherwise. Note that all the tasks share the Image-Structure model. Except the UI Object Detection task, they also share the Question-Answer model. We do not introduce a special task token. Instead, $Q$ is served as the natural language task indicator for signaling the model to produce different answers for each task. $Q$ also carries the actual task specifics for the grounding task to find the object on the UI.

These five tasks are very different in nature. UI object detection only takes image input and generates UI objects. Command grounding leverages all the input modalities and outputs an object reference. Tappability, UI summarization and widget captioning share the same input and output modalities but are fundamentally different. Tappability is concerned with the usability of a specific UI object. In contrast, UI summarization and widget captioning generate functional descriptions about UIs, with the former focusing on the entire screen and the latter concentrating on a specific object. These heterogeneous tasks pose challenges for multi-task learning.

## 4 Model Architecture

As shown in Figure 1, the architecture of VUT includes two Transformer models, i.e., the Image-Structure model and the Question-Answer model, and three output heads for three types of responses that accomplish 5 tasks.

### 4.1 Image-Structure Transformer

The Image-Structure Transformer is a two-modal model, which takes an image and the corresponding view hierarchy, and outputs the hidden representation of the image and each node in the view hierarchy. For the image modality, we compute the content embedding, $C_s$, and its positional encoding, $P_s$, according to DETR (Carion et al., 2020): $C_s = \text{Dense}(\text{ResNet}(S))$ and $P_s = \text{PE}(S_{mask})$. $S_{mask}$ is the binary non-padding mask of the image $S$. We omit details such as tensor reshaping and broadcasting in our discussions here. $C_s \in \mathbb{R}^{M \times D}$ and $P_s \in \mathbb{R}^{M \times D}$ where $M$ is the flattened size of the feature map after ResNet and $D$ is the dimension of the representation.

For the view hierarchy modality, when $V$ is absent in the object detection task (Section 3.1), the content embedding for the modality, $C_v$, is all zeros, and the positional encoding, $P_v$, is a learned embedding vector for each query position. When the view hierarchy is present in the input, each object in the view hierarchy tree, $V$, is embedded based on the set of attributes it possesses. The discrete attributes such as type, clickability, and text content are embedded separately to the same dimension and then combined via addition to form the content embedding of each element, $C_v$. When there are multiple tokens in text content, max pooling is used to acquire a single fixed-size representation of text content.

The positional encoding of each object, $P_v$, is computed based on its bounding box [top, left, right, bottom] and DOM positions [pre-order idx, post-order idx, depth]. Each is treated as a continuous

vector position that is encoded via learnable Fourier representation (Li et al., 2021b) and then summed together to form the positional encoding of the object. When $V$ is not present in the input, the position encoding is simply a learned embedding for each index position in the input. The final embedding of the view hierarchy modality includes two tensors: $C_v \in \mathbb{R}^{N \times D}$ and $P_v \in \mathbb{R}^{N \times D}$.

$P_s$ and $P_v$ are positional encoding within each modality. Because the embeddings from the two modalities will jointly participate in the self attention of the Transformer encoder, it is important to make their positional encoding global instead of local to each modality. To this end, we add a learnable modal embedding to each of these modality-specific positional encoding: $P'_s = P_s + E_s$ and $P'_v = P_v + E_v$, where $E_s \in \mathbb{R}^{1 \times D}$ and $E_v \in \mathbb{R}^{1 \times D}$ are the learnable embeddings for the image and view hierarchy modality respectively.

When the task is with respect to a specific object on the screen (Section 3.2 and 3.5), $T$ is passed to the model so that it can pay more attention to the object. To achieve this unique need of UI tasks, we modulate the content embedding of both modalities, $C_s$ and $C_v$ using a *focus map* $\alpha_s$ produced from $T$ as follows.

$$\alpha_s = \text{Softmax}(\text{Flatten}(\text{RegionMask}(T_{bbx}))\beta + \tau)M$$
$$C'_s = \alpha_s \otimes C_s$$
$$(6)$$

RegionMask$(\cdot)$ creates a 2D binary mask from the bounding box of the target object $T_{bbx}$ where it is all ones inside the the box and all zeros outside. Flatten$(\cdot)$ flattens the 2D mask to 1D. $\beta$ and $\tau$ are the hyperparameters that regulate how much the model should focus on the target object versus the rest of the image. $\otimes$ denotes element-wise multiplication. We perform a similar modulation for the target object in $V$.

$$\alpha_v = \text{Softmax}(\text{OneHot}(T_{idx})\beta + \tau)N$$
$$C'_v = \alpha_v \otimes C_v$$
$$(7)$$

OneHot$(\cdot)$ produces a one-hot vector from the index of the target object, $T_{idx}$, in the view hierarchy. We then concatenate the embeddings of the two modalities along the first dimension to form the input the Transformer encoder: $C = \text{Concat}[C'_s; C'_v]$ and $P = \text{Concat}[P'_s; P'_v]$, where $C \in \mathbb{R}^{(M+N) \times D}$ and $P \in \mathbb{R}^{(M+N) \times D}$ are the final content embedding and positional encoding respectively, which are fed to a multi-layer Transformer encoder: $H = \text{Transformer\_Encoder}(C, P)$, where the hidden representation $H \in \mathbb{R}^{(M+N) \times D}$. We then split $H$ for the hidden representation for each modality: $H_s = H[: M]$ and $H_v = H[M :]$ while result in the hidden representations for each modality: $H_s \in \mathbb{R}^{M \times D}$ and $H_v \in \mathbb{R}^{N \times D}$.

## 4.2 QUESTION-ANSWER TRANSFORMER

The Question-Answer Transformer is a language model that encodes the question $Q$ and decodes the answer $A$ and produces the hidden representation for the grounding task. The input of the model, $X = x_{1:t}$, where $t$ is the length of the sequence, is either just the token sequence of $Q$ for the grounding task (Section 3.4), or the concatenation of $Q$ and the decoded answer $A'$ when a language answer is to be generated. During training with teaching forcing, $A' = A$. During auto-regressive inference, $A'$ is the predicted token sequence up to a step.

$$g_{1:t} = \text{Transformer\_Decoder}(E(x_{1:t}), PE(1:t); H_s, H_v)$$
$$(8)$$

where $x_i$ is the $i$th token in the sequence ($1 \le i \le t$), and $E(\cdot)$ and $PE(\cdot)$ compute the content embedding and the positional encoding of each token in the sequence. $H_s$ and $H_v$ are accessed via Transformer encoder-decoder attention. The sequence of hidden states, $g_{1:t}$, $g_i \in \mathbb{R}^D$, are used for predicting the next token for generating an answer or for retrieving a target UI object in the view hierarchy for the grounding task.

### 4.3 OUTPUT HEADS

**Object Detection Head**: For the UI Object Detection task (Section 3.1), we borrow the object output layer from DETR (Carion et al., 2020), using $H_v$ as the input the layer: $Y_{type} = H_v W_{type}$ and $Y_{bbx} = \phi(H_v, \theta_{bbx}) W_{bbx}$ where $W_{type} \in \mathbb{R}^{N \times 22}$ is the linear projection to output the object type logits. An additional PADDING type is included on top of the original 21 UI object classes. $\phi(\cdot)$ is the multi-layer perceptron parametized by $\theta_{bbx}$ and $W_{bbx} \in \mathbb{R}^{N \times 4}$ is the linear projection for generating the coordinates. The logits $Y_{type} \in \mathbb{R}^{N \times 22}$ and the coordinates $Y_{bbx} \in \mathbb{R}^{N \times 4}$ are for both generating object predictions and computing optimal compound loss using Hungarian Matching during training Carion et al. (2020).

**Text Head**: For the three tasks that require a text response, $A$, (Section 3.2,3.3 and 3.5), we apply a softmax layer on top of the decoder hidden state, $g_{1:t}$ (Equation 8), to generate each answer token.

$$a_i = \arg\max(\text{Softmax}(g_{|Q|+i-1} W_{txt})) \tag{9}$$

where $a_i$ is the $i$th token in the answer sequence $A$, and $|Q|$ is the length of the question. $W_{txt} \in \mathbb{R}^{D \times |\text{vocab}|}$ is the learnable weights and $|\text{vocab}|$ is the vocabulary size. For the three tasks, we optimize the model for the cross-entropy loss over the predicted and ground-truth answer token sequences.

**Pointer Head**: For the grounding task (Section 3.4), we use the last hidden state from the Transformer decoder as a "pointer" Vinyals et al. (2015) to match against all the objects in the UI based on their hidden representations, using dot product similarity (Equation 10).

$$\hat{t} = \arg\max_{1 \le j \le N}(\text{Softmax}(g_{|Q|} W_{ptr} \cdot h_j) \tag{10}$$

where $h_j$ is the $j$th row in $H_v$ that is the hidden representation of the $j$th object in the view hierarchy. $W_{ptr} \in \mathbb{R}^{D \times D}$ is the learnable projection and $g_{|Q|}$ is the last hidden state from the decoder (Equation 8), which is able to access the entire question (command) sequence, $Q$, via the decoder self attention. Compared to previous UI ground work (Li et al., 2020a), we used the last hidden state as the "pointer" instead of embedding pooling of a bag of word in a span. We optimize the model by minimizing the cross-entropy loss between the predicted and the ground-truth object index.

## 5 EXPERIMENTS

### 5.1 DATASETS

For the UI Object Detection task, we use RICO, a public corpus of mobile user interfaces (Deka et al., 2017) that contains 64,462 unique Android screens from 9,362 different apps. Each screen comes with an RGB screenshot and a corresponding view hierarchy. A view hierarchy is a tree structure of nodes with 21 unique types, which we consolidated from the Android View class attributes in the original dataset (see Appendix D). Plus the special PADDING type, there are 22 types to be predicted. A view hierarchy has a maximum of 128 nodes in our dataset.

For the Widget Captioning task, we used a public dataset (Li et al., 2020b), which includes more than 200k human annotations for over 46k unique UI objects from 17k RICO screens. For Screen Summarization, we used a public dataset (Wang et al., 2021) that consists of 112k human created summaries for 22,301 unique Android screens from RICO.

We created the Language Grounding dataset, which has 10k human annotations for operating UI objects of 1432 unique screens from 26 Android build-in apps like Settings. A human rater generates commands such as "Click the button below battery info", and the maximum length of a command phrase is 20 words. The Tappability Prediction dataset includes tappability annotations for more than 18,669 UI elements from 3,218 Android screens. In the data collection, given a target UI element highlighted on a screen, 5 different human raters are asked to answer yes or no for whether the target object looks tappable to them. We use the majority voting to determine the label of each element.

We split each dataset into training, validation and test sets (Table 2), and ensure there is no overlap of apps (or screens) between a training set and any of the test sets of different tasks. This is important

| Dataset | Train | Validation | Test |
|---|---|---|---|
| UI Object Detection (Deka et al., 2017) | 54,611 | 2,518 | 2,627 |
| Widget Captioning (Li et al., 2020c) | 39,951 | 3,436 | 3,531 |
| Screen Summarization (Wang et al., 2021) | 17,569 | 2,298 | 2,434 |
| Language Command Grounding | 7,822 | 1,024 | 987 |
| Tappability Prediction | 14,783 | 1,854 | 2,029 |

Table 2: Datasets.

because in the multi-task learning condition, VUT learns from all the training sets. Thus the union of apps and screens across all the training sets should not overlap any of the test set. We also ensure our test datasets are the same with the released benchmarks for widget captioning and screen summariztion so that the results are comparable with reported SOTAs. We unify these heterogeneous datasets to follow the same feature taxonomies based on our problem formulation so that they can be consumed by the same model.

## 5.2 RESULTS

### 5.2.1 COMPARING VUT WITH DETR & CENTERNET FOR OBJECT DETECTION

In this experiment, we compare VUT with two benchmark models for UI Object Detection. The standard DETR architecture (Carion et al., 2020) uses a 6-layer Transformer encoder and a 6-layer Transformer decoder. To have a similar number of parameters in the model, we let VUT Image-Structure transformer to use a 12-layer Transformer encoder in this experiment. CenterNet (Duan et al., 2019) has been a popular choice for object detection, which achieved SOTA results on natural images. In this experiment, we let CenterNet use ResNet-101 as its backbone to reach a similar parameter size. Table 3 shows that VUT's Image-Structure model clearly outperforms DETR and CenterNet on the UI Object Detection dataset. Previously, Carion et al. (2020) found more encoding layers can lead to better accuracy. It is worth noting that VUT's Image-Structure model uses an encoder-only architecture that achieves better accuracy.

| Model | #Params | AP | $AP_{50}$ | $AP_{75}$ | $AP_{small}$ | $AP_{medium}$ | $AP_{large}$ |
|---|---|---|---|---|---|---|---|
| CenterNet | 49M | 31.9 | 44.3 | 33.0 | 1.2 | 12.6 | 33.0 |
| $DETR_{6+6}$ | 50M | 37.8 | 49.1 | 39.6 | 1.8 | 21.1 | 38.1 |
| $VUT_{12}$ | 48M | **39.3** | **50.1** | **40.9** | **3.3** | **21.6** | **39.6** |

Table 3: Comparison of three models for UI Object Detection on the validation dataset.

Note that these results can not be compared directly with those previously reported Zhang et al. (2021). Our task is more challenging for detecting 21 different UI object types including several container elements, instead of 12 objects in previous work. In addition, previous work used a dataset that is manually labeled by human and also employed heavy post-process to improve the accuracy.

### 5.2.2 COMPARING MULTI-TASK WITH SINGLE TASK LEARNING

For each task, we report its accuracy when it is learned alone and with other tasks jointly. We also show the benchmark results on these datasets reported by previous works for comparison, when they are available. In this experiment, we use a 6-layer Image-Structure model and a 6-layer Question-Answer model. Each task head and model parts are used only for batches that are specific for each task during training. For Widget Captioning, Screen Summarization and Tappability, text head (Equation 9) is used for generating answers. For Language Command Grounding, the grounding head (Equation 10) is used instead. See Appendix B.1 for training schedule details.

As shown in Table 4, 5, 6 and 7, multi-task learning, though more challenging than single-task learning, can often perform on par with single-task learning. It consistently outperforms single-task learning for most configurations and metrics. There is a decrease of accuracy for the Grounding task when text-generation related tasks are involved. We suspect that the grounding task, which relies on the last hidden state of the Question-Answer model, probably competes with the three text-generation tasks by "pulling" the hidden representations of the Question-Answer model towards different

| Configurations | BLEU-1 | BLEU-2 | BLEU-3 | BLEU-4 | ROUGE | CIDEr |
|---|---|---|---|---|---|---|
| SOTA, Li et al. (2020c) | 44.9 | 32.2 | - | - | 44.7 | 97.0 |
| Widget Captioning alone | 45.8 | 30.2 | 19.6 | 12.9 | 46.0 | 94.8 |
| Widget Caption + Object Detection | 46.7 | 31.6 | 21.9 | 15.0 | 45.9 | 98.3 |
| 4 tasks (without Object Detection) | 43.3 | 28.5 | 18.7 | 14.0 | 44.0 | 88.9 |
| All 5 tasks | **47.0** | **32.3** | **22.7** | **16.3** | **46.8** | **99.3** |

Table 4: Results for the Widget Captioning task

| Configurations | BLEU-1 | BLEU-2 | BLEU-3 | BLEU-4 | ROUGE | CIDEr |
|---|---|---|---|---|---|---|
| SOTA (Wang et al., 2021) | 65.5 | 45.8 | 32.4 | 25.1 | 48.6 | 61.3 |
| Screen Summarization alone | 68.7 | 49.4 | 31.6 | 19.4 | 53.8 | 64.3 |
| Summarization + Object Detection | **68.9** | **50.8** | **33.5** | **21.4** | **54.9** | **65.6** |
| 4 tasks (without Object Detection) | 68.2 | 49.4 | 32.2 | 20.2 | 53.5 | 56.8 |
| All 5 tasks | 67.7 | 49.2 | 32.1 | 20.1 | 53.9 | 65.1 |

Table 5: Results of the Screen Summarization task.

| Configurations | Ground accuracy (%) |
|---|---|
| Command Grounding without image input | 68.5 |
| Command Grounding alone | 75.5 |
| Command Grounding + Object Detection | **82.1** |
| 4 tasks (without Object Detection) | 77.3 |
| All 5 tasks | 80.8 |

Table 6: Results for the Language Command Grounding task.

| Configurations | Precision (%) | Recall (%) | F1 (%) |
|---|---|---|---|
| Tappability alone | **91.9** | 81.9 | 86.6 |
| Tappability + Object Detection | 91.0 | 84.6 | 87.7 |
| 4 tasks (without Object Detection) | 86.5 | 84.8 | 86.7 |
| All 5 tasks | 90.1 | **86.5** | **88.3** |

Table 7: Results for the Tappability task.

| Configurations | AP | $AP_{50}$ | $AP_{75}$ |
|---|---|---|---|
| Object Detection alone | 37.0 | 47.6 | 38.8 |
| Widget Captioning + Object Detection | 36.6 | 47.8 | 38.5 |
| Screen Summarization + Object Detection | 36.3 | 47.8 | 38.3 |
| Command Grounding + Object Detection | 36.9 | 48.4 | 38.8 |
| Tappability Prediction + Object Detection | 37.6 | 48.4 | 39.5 |
| All 5 tasks | 35.2 | 46.8 | 36.8 |

Table 8: Accuracy for the UI Object Detection task when different tasks are jointly learned.

directions. One phenomenon that we consistently observed is that having the Object Detection task in multi-task learning often outperforms the configuration without involving Object Detection. For the Object Detection task itself, we found there is a drop of accuracy when batch-alteration for multi-task learning starts. Yet, it mostly recovers its accuracy (see Table 8). These experiments show that instead of treating Object Detection as a standalone pretraining task, it is feasible for it to be part of the multi-task learning where VUT achieves all the tasks through a single model.

## 6 CONCLUSION

We present VUT, a multi-modal Transformer for multi-task modeling of user interfaces. Our model takes in three types of data, i.e., UI screenshot images, view hierarchy structures, and natural language questions. Our experiments based on 5 datasets show that VUT achieves five types of UI tasks simultaneously, and show the promise of providing unified modeling for the user interface domain.

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

## A  QUESTIONS

As discussed in the paper, there are three types of question input, $Q$, fed into the Question-Answer model: commands, questions for yes-or-no responses, and questions for open-ended answers. When VUT is joint-learning all the tasks, it is required to recognize all the questions for different types of tasks. For questions that expect a textual response, e.g., Widget Captioning, Screen Summarization, and Tappability Prediction, VUT also needs to generate an answer corresponding to each type of question.

### A.1  COMMANDS

For the Language Command Grounding task, commands that refer to a specific object in the screen are fed to the model by which the model is trained to locate the referred object. Example commands are shown below. These commands are created by human annotators for a target UI object shown on a screen. A human annotator is asked to come up with different commands referring to each highlighted target object.

- *Click on the notification bar above the status option.*
- *Press on the back arrow button.*
- *Select the icon above the clock option.*
- *Swipe down the notification bar.*

### A.2  QUESTIONS FOR YES-OR-NO ANSWERS

For the Tappability Prediction task, we generate synthetic Yes-or-No questions based on the following regular expression pattern. The model is trained to decode *yes* or *no* as the answer to the question.

```
Is the [object|element|widget|control] [clickable|tappable]?
```

The question examples that are generated based on the regular expression are the following.

- *Is the object tappable?*
- *Is the widget clickable?*
- *Is the element tappable?*

### A.3 QUESTIONS FOR OPEN-ENDED ANSWERS

For Screen Summarization summary and captioning tasks, the model will need to generate an open-ended answer. We use the following regular expressions to generate questions for these tasks. VUT is trained to decode a screen summary or a widget caption following the question.

```
What is the [summary|description] of the [screen|UI]?
```

```
What is the [caption|description] of the [object|element|widget|control]?
```

Below are the question examples generated from the above regular expressions.

- *What is the summary of the screen?*
- *What is the description of the UI?*
- *What is the caption of the widget?*
- *What is the description of the object?*

## B  MODEL & TRAINING DETAILS

For the comparison with DETR and CenterNet on the UI Object Detection task, VUT uses a 12-layer Transformer encoder as the Image-Structure model that amounts to 48M trainable parameters, which is fewer than 50M trainable parameters of DETR with a 6-layer encoder and a 6-layer decoder, and 49M parameters of CenterNet. Image Embedder (Figure 1) is the ResNet backbone. For the remaining experiments, VUT uses a 6-layer Transformer encoder for the Image-Structure model, and a 6-layer Transformer decoder for the Question-Answer model. When all the tasks are jointly trained, there are 64M parameters. Task-specific heads and word piece embeddings and projections are the main contributors to the growth of the parameter size. When only a subset of these tasks is involved in the training, e.g., Widget Captioning + Object Detection, there will be fewer trainable parameters involved because only part of the full model is in use. All the VUT variants use the following configurations: #Attention_Heads=8, Hidden_Dimension=256, Transformer_MLP_Dimension=2048, Transformer_QKV_Dimension=256.

Note that in our Image-Structure model, positional encoding is added to the input of each layer of the Transformer. This is in contrast to our Question-Answer model where the positional encoding of each token is only added to the input of the first layer. We use the learned embedding for positional encoding in the Question-Answer model. During training, we use 10% for both the attention and the MLP dropout in the Questions-Answer Transformer, and we also apply 10% dropout on the encodings from the Image-Structure model before cross attention. During the 5-task joint learning, the attention and the MLP dropout rates are 10% for the Image-Structure Transformer and the encoder-decoder cross-attention dropout rate is 20%. During auto-regressive decoding for interference, the maximum decoding length is 30 that covers the total length of a question and an answer.

We use the same method as BERT for tokenizing phrases into a sequence of word pieces[3], which results in a vocabulary size of 28,536. The maximum size for a screenshot image is $1080 \times 1080$. We randomly resize each image for image augmentation. The maximum number of UI objects and containers on each screen is capped to 128. We implement VUT based in JAX[4], a library for machine learning. We train each VUT model with a batch size of 64 screens/examples, which the training is parallelized across 64 TPU v3 cores.

### B.1 TRAINING SCHEDULE DETAILS

Because the UI Object Detection task requires many more iterations to converge than other tasks, we start our multi-task learning by training VUT for the UI Object Detection task, and then training VUT jointly for multiple tasks by alternating batches from each dataset. This learning strategy is sensible because by learning from the UI Object task, the model can learn useful information about

---

[3]https://www.tensorflow.org/tutorials/tensorflow_text/subwords_tokenizer#overview

[4]https://github.com/google/jax

how to encode the screen pixels. As it is consistently shown in our experiments, joint learning that involves Object Detection can often boost the learning of the other four tasks.

We start the training by learning the Object Detection task that takes 300k steps, using DETR's default training schedule (Carion et al., 2020). For joint-learning of all the 5 tasks, we sample batches from the Object Detection, Widget Captioning, Scren Summarization, Command Grounding and Tappability datasets with the weights $[15, 10, 10, 20, 1]$ and the model is trained with 100k steps. The learning rate is 1e-4 and decayed to 1e-5 after 50k steps.

For individual task training, we train each model until it converges with a batch size of 64. For the UI Object Detection task, we train the model with the most default setup of DETR with 6-layer encoder and 6-layer decoder and 8-head attention and 256 hidden size, using 300k steps. The learning rate schedule includes one learning rate decay from 1e-4 to 1e-5 at the 200k steps. The Widget Captioning task uses a 6-layer Image-Structure model and a 6-layer Question-Answer model plus the Text head. Similarly, the model was trained 45k steps to converge. For the Screen Summarization task, we used the same model configuration as Widget Captioning, and the model converges at 50k steps. The Language Command Grounding task uses a similar model setup as the Widget Captioning and the Screen Summarization tasks, except that it uses the Grounding Head instead of the Text Head. It took the model 27k steps to converge. For training the model for each of these tasks, we decay the learning rate once from 1e-4 to 1e-5 at 15k steps. For learning the Tappability Prediction task alone, we used the same model setup as the two text related tasks (Summarization and Captioning). We found the model is very prone to overfitting in spite of using a large dropout rate. So we train the model with early stopping.

For Object Detection + another individual task, we sampled batches from the two datasets with weights similar to the ones used for 5 task joint training. For Widget Captioning, Screen Summary and Grounding, the models are trained to converge with 35k, 50k and 50k steps, with 1e-4 initial learning rate which decays to 1e-5 at 25k, 15k and 25k steps, respectively. For the Tappability task, we use a initial learning rate of 1e-5 and train the model with early stopping.

For 4 task joint training (without Object Detection), we sampled batches from the 4 tasks (Widget Captioning, Screen Summary, Grounding, Tappability) with the weights $[5, 5, 10, 1]$. We trained the model with 150k steps and batch size 64. The learning rate schedule includes one learning rate decay from 1e-4 to 1e-5 at the 50k steps.

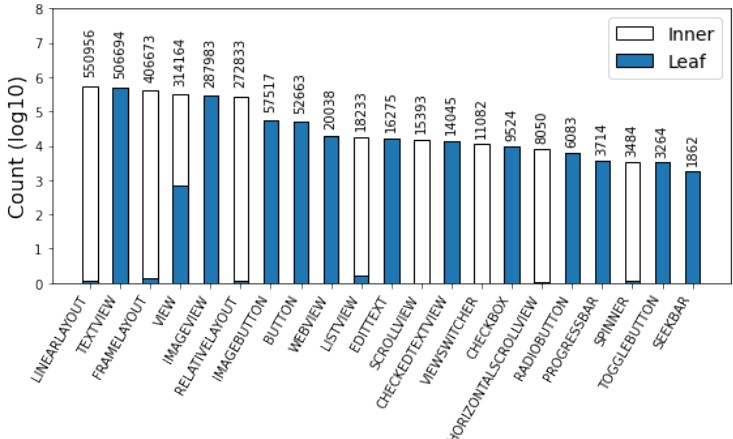

Figure 2: The distribution of the 21 UI elements at the $log_{10}$ scale. The proportion that each type is used as an inner versus a leaf node is shown within each bar.

## C    ABLATION STUDY ON FOCUS OBJECT

For Widget Captioning and Tappability task, VUT predicts the caption or the tappability answer of a given object on the screen—the *focus object*. To inform the Image-Structure Transformer the location of the given object, we apply a focus map to the Image Embeder outputs and the embeddings

of the view hierarchy objects so that the focus object has more weight on its pixel area and object embedding than the others (see Equation 6 and 7). Specifically, we used $\beta = 2.0$ and $\tau = -1.0$ in our experiments. To show the effectiveness of the focus modulation, we conduct an ablation study using the Widget Captioning model with three settings: 1) applying the focus map to both the image and structure modalities, 2) applying the focus map to the structure modality only, and 3) concatenating an embedding of $\{0, 1\}$ to the object embedding for the structure modality with 1 indicating the focus object.

As shown in Table 9, the model performs the best when focus map is used on both the image and the structure modalities, indicating the necessity and effectiveness to inform the model the focus object location in the pixels and the object list. Using the focus map only on the object embedding is not sufficient because the Image-Structure model needs to learn from the data to find the corresponding pixel areas of the focus object. Yet, using $\{0, 1\}$ embedding is the least effective.

| Configurations | BLEU-1 | BLEU-2 | ROUGE | CIDEr |
|---|---|---|---|---|
| $\{0, 1\}$ embedding on the structure modality | 39.7 | 26.7 | 38.6 | 80.7 |
| Focus map on the structure modality only | 41.0 | 27.3 | 39.8 | 84.9 |
| Focus map on both the image and structure modality | **46.4** | **31.4** | **45.1** | **98.5** |

Table 9: Ablation Study Results for Focus Weight

## D    UI OBJECT TYPES

We process the RICO dataset for the UI Object Detection task. As discussed in the paper, among the 64,462 screens of the original dataset, we particularly use those verified by human raters for validation and test datasets. For each element on the screen, we extract its attributes such as its UI object type, its bounding box position on the screen, whether it is clickable or enabled. We exclude all the elements that are marked as invisible as they have no correspondence with pixels on the screen. There are many custom object types in the dataset and many of them inherit from common Android widget types[5]. We consolidate rare object types (such as TABWIDGET or VIDEOVIEW) to their closest ancestor type that are in the common widget library. With the consolidation, there are 21 UI object types in the dataset (Figure 2), which has a long tail distribution. As we can see from the distribution, some elements are dedicated as leaves in a view hierarchy, e.g., ImageButton or RadioButton, and some elements are primarily for determining the layout or as non-terminal nodes, e.g., LinearLayout. There are cases that a type is used for both leaves and non-leaves, e.g., View. The View type often catches UI elements that cannot be easily classified into a specific type. Together, these make UI Object Detection a challenging task.

## E    MODEL SIZE AND INFERENCE TIME COMPARISON

In this section, we compare model size and inference time of the 5 tasks when they are learned jointly and individually. We can see the joint model has the largest parameter size, as it includes all the task heads. However, the size of the 5-task joint model is substantially smaller than the total size of each individual single-task model (Table 10). The inference time of the joint model for each task is similar that of the task-specific model (Table 11), as only the part that is required by the task is activated in the joint model during inference. Altogether, VUT can achieve the five tasks with little inference time overhead and a substantially small model footprint.

## F    EXAMPLES OF PREDICTION RESULTS

We here show examples of predictions versus ground-truth for each task, on the test data, as achieved by a single model of VUT, when it learns all the tasks jointly.

---

[5]https://developer.android.com/reference/android/widget/package-summary

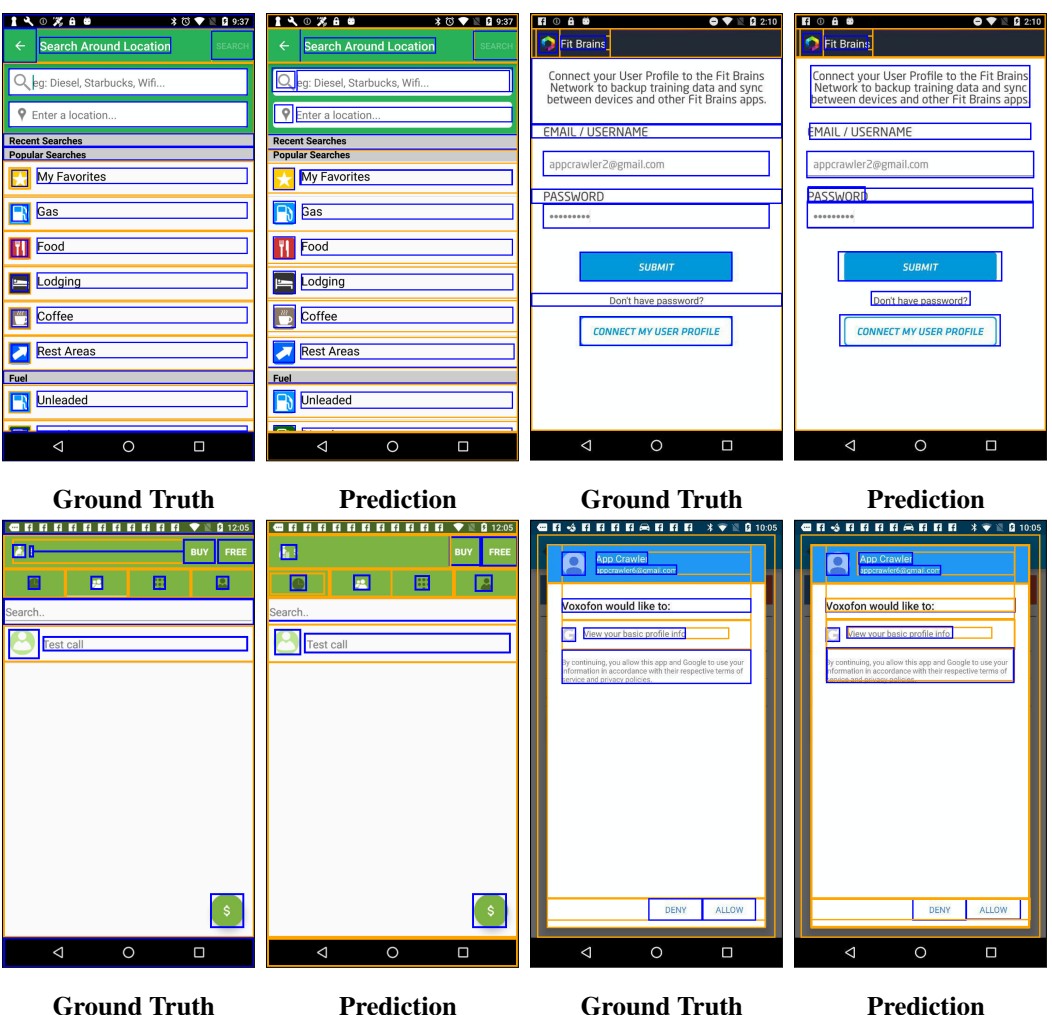

Figure 11: Examples for the UI Object Detection task. In the ground-truth screens, the bounding boxes of inner objects are highlighted in orange and those of leaf objects are shown in blue. In the predictions, we render the bounding boxes of predicted objects as inner (orange) versus leaf (blue) based on the dominant use of the predicted object type, according to Figure 2.

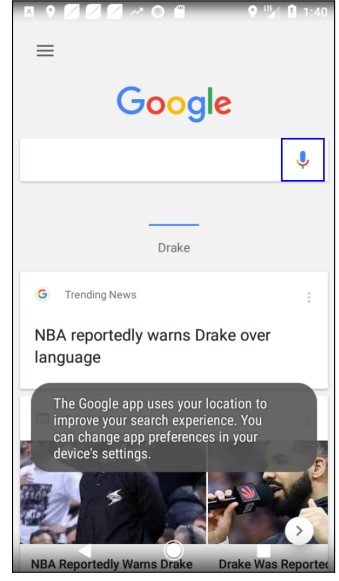 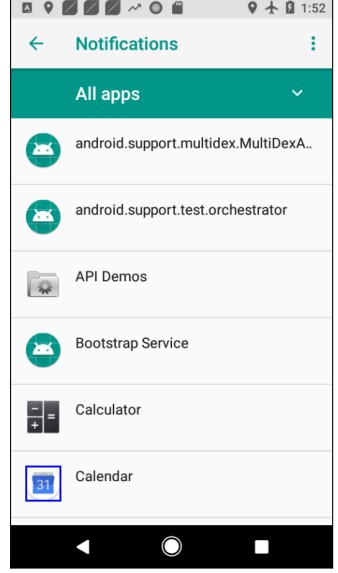 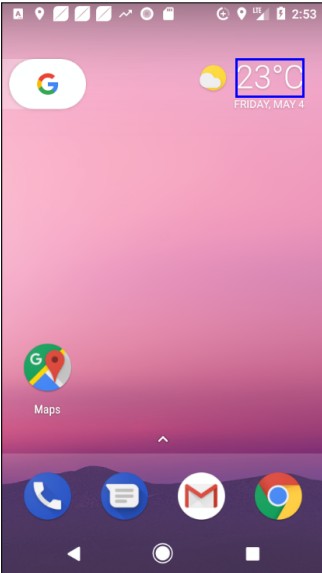

**Command**: *"tap on the voice icon"*

**Command**: *"press on the icon below calculator icon"*

**Command**: *"select the weather text which is below the notification bar"*

Figure 15: Examples for the Language Command Grounding task. The object located by the model is highlighted with a blue bounding box in each screenshot.

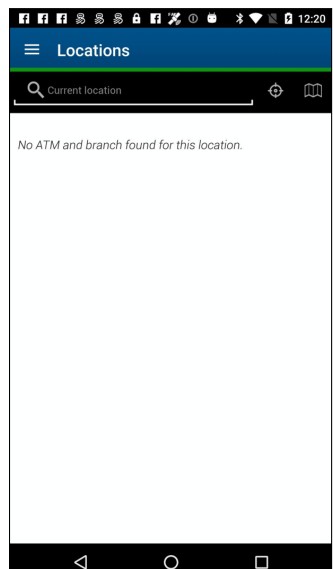 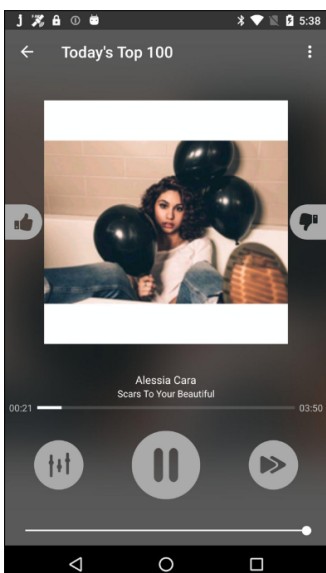 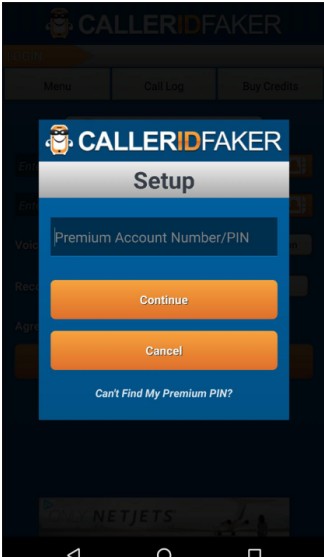

**Prediction**: *"search bar to search for the location"*
**Reference**: *"page displaying a search box in the app"*

**Prediction**: *"page displaying music track in music app"*
**Reference**: *"screen shows music playing on an app"*

**Prediction**: *"pop-up showing to create an account"*
**Reference**: *"pop-up displaying to setup the account details"*

Figure 19: Examples for the Screen Summarization task. We here display one of the 5 references (ground-truth summaries) created by human annotators for each screen.

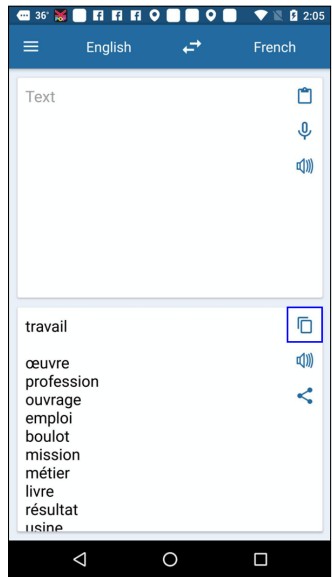
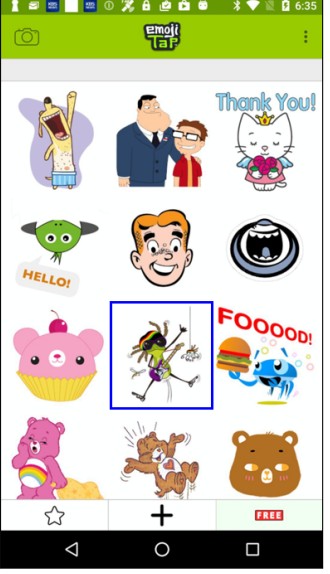
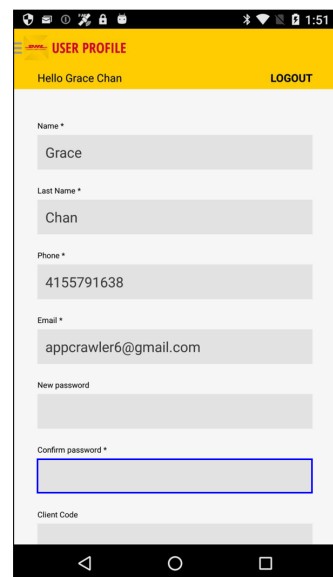

**Prediction**: *"copy text"*
**Reference**: *"copy to clipboard option"*

**Prediction**: *"select the emoji"*
**Reference**: *"select guitar lizard"*

**Prediction**: *"enter password"*
**Reference**: *"input confirm password"*

Figure 23: Examples for the Widget Captioning task. The target element is highlighted via a blue bounding box. We here show one of the three references (ground-truth captions) created by human annotators for each target element.

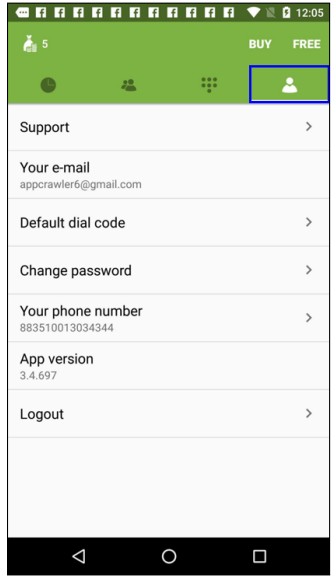
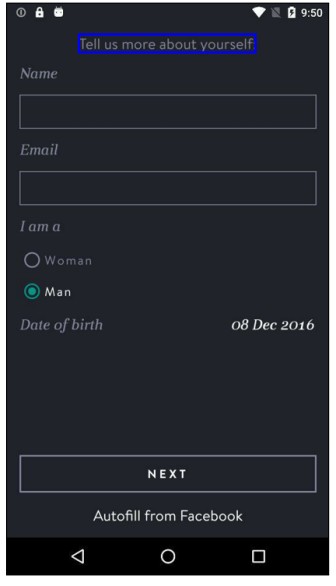
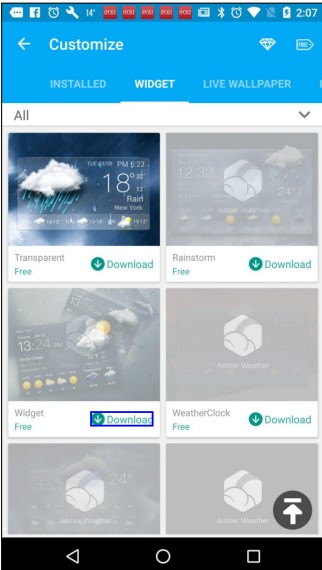

**Prediction**: *"yes"*
**Ground Truth**: *"yes"*

**Prediction**: *"no"*
**Ground Truth**: *"no"*

**Prediction**: *"yes"*
**Ground Truth**: *"yes"*

Figure 27: Examples for the Tappability Prediction task. The questioned element is highlighted with a blue bounding box.

| Task | 5-Task Joint Model | Single-Task Model |
|------|:---:|:---:|
| Object Detection | 63.6 | 39.5 |
| Widget Captioning | 63.6 | 56.2 |
| Screen Summarization | 63.6 | 56.2 |
| Command Grounding | 63.6 | 63.5 |
| Tappability | 63.6 | 56.2 |
| 5 tasks | 63.6 | 271.6 |

Table 10: Parameter size (millions) comparison between the 5-task joint model and single task model. The 5-task joint model performs the five tasks simultaneously, which otherwise require five separate models that amount to a much larger model footprint.

| Task | 5-Task Joint Model | Single-Task Model |
|------|:---:|:---:|
| Object Detection | 12.21 | 11.59 |
| Widget Captioning | 30.02 | 30.00 |
| Screen Summarization | 43.68 | 42.97 |
| Command Grounding | 70.53 | 71.29 |
| Tappability | 51.55 | 51.09 |

Table 11: Inference time (ms) comparison between the 5-task joint model and single task model for each task. The time is calculated by averaging the inference times on the test set. There is little time overhead for the joint model to perform each task, compared to each task-specific model.

