# OpenReview forum: "VUT: Versatile UI Transformer for Multimodal Multi-Task User Interface Modeling "
_ICLR.cc/2022/Conference — ICLR 2022 Submitted_

### Official Review · Reviewer_69JK · 2021-11-01

**Correctness:** 3
**Technical Novelty And Significance:** 2
**Empirical Novelty And Significance:** 2
**Recommendation:** 5
**Confidence:** 4

**Main Review:**

**Strengths:**

1. Overall, the formulation of the problem and various design choices (global positional encoding, focus map, pointer head for the grounding task) make sense. The authors also seem to have carefully designed their experiments. For instance, ensuring that comparisons are fair (same number of parameters, etc) for the object detection task.
2. Given the nature of the problem statement (with multiple tasks, inputs and outputs), the authors have done a good job in explaining each of them properly. Overall, I thought the paper was easy to read and understand.
3. The authors provide several reasonable insights that might be relevant to the user interface modeling community — using object detection as part of multi-task learning instead of standalone pre-training task, design choices to create a single unified architecture for all the tasks, multi-task learning outperforming single-task learning etc.

**Weaknesses**

1. I understand that a single model is helpful for multiple UI tasks, but I wonder if this approach is scalable beyond the 5 tasks and 5 modalities mentioned. The authors comment that the model architecture is designed to remain stable for a growing set of tasks, but this seems to be under the assumption that the input and output modalities would remain constant. A discussion on how to add new tasks to the same framework might help, or a discussion on why the current framework is enough.
2. One of the important motivations of multi-modal multi-task learning mentioned was to achieve better or on-par performance with a single model (and supposedly fewer computations) which is crucial for devices with limited computing resources. But a direct head to head comparison for the computational cost of a multi-task model and individual models is not provided. Are there any overheads/disadvantages because of multi-task learning (Like a larger model size, inference time for individual tasks etc)? The model sizes are varied across experiments to achieve on par performance which makes the comparison of the computational cost not so obvious. Such a comparison would highlight the advantages of the multi-task model and would be helpful for the relevant audience.
3. I also believe that the downstream tasks are also somewhat similar (language command grounding, tappability, UI object detection, UI summarization, widget captioning). It is therefore not surprising that multi-task learning should help these tasks.
4. There is no comparison provided with any baseline for 2/5 tasks (Language grounding, Tappability)
5. The novelty in the paper is somewhat lacking. The techniques used in the paper (multi-branch transformer, pointing mechanism, cross-modal attention, global positional encodings, etc) have been shown to work in the past for image-text tasks [1, 2]. The paper is taking all the lessons from past works and applying it to a new domain.

**Clarifications:**

1. The training procedure mentioned in section 5.2.2 talks about joint training but the procedure followed for training for individual tasks or a subset of tasks is not described in detail. Were the same hyperparameters used for all configurations?

**Updates after rebuttal period**
The authors addressed some of the concerns -- showing inference time, model size and a discussion about training details and hyperparameters in the appendix. However, I am not convinced that the paper presents new insights that are relevant for the broader ICLR community. Concerns around novelty and the multi-task setup was also raised by another reviewer (e7Hg). Responses to reviewer e7Hg also aren't convincing. If the tasks are not similar, and the learning objectives are not aligned, then the motivation for multi-task learning is solely for reducing memory footprint and computational cost. I believe that this contribution isn't enough for me to recommend acceptance.   Thus, I am going to stick to my original rating.

[1] R. Hu, A. Singh, T. Darrell, M. Rohrbach, *Iterative Answer Prediction with Pointer-Augmented Multimodal Transformers for TextVQA*. in CVPR, 2020

[2] J. Lu, V. Goswami, M. Rohrbach, D. Parikh, S. Lee, 12-in-1: Multi-Task Vision and Language Representation Learning. in CVPR, 2020

**Summary Of The Paper:**

The paper proposes an architecture for graphical user interfaces which involve multi-modal inputs (UI screenshots, Hierarchy structures, Natural Language) and multi-task learning (UI Object Detection, Widget Captioning, Screen summarization, Language grounding, and Tappability).The proposed architecture consists of seperate transformer blocks to encode image and text modalities. The two transformer blocks attends to each other to produce multi-modal outputs, which is then used for downstream tasks. The authors that with the proposed architecture, training on all tasks simultaneously is better than training on individual task alone.

**Summary Of The Review:**

Overall, the paper showed how multi-task learning can improve performance on several downstream UI modeling tasks. These insights might be relevant to the UI modeling community. The authors use several popular ideas in the field of multi-modal, multi-task learning and showed it's effectiveness on multiple downstream tasks. The paper is well written, and the ideas are clearly presented but it lacks in technical novelty. Additionally, the paper lacks in a few other areas -- comparison to single-task models with respect to computational cost, inference time etc, the downstream tasks are similar but the performance gains aren't significant. Due to these reasons, I think the paper is marginally below the acceptance threshold.

---

> ### Author Response · Authors · 2021-11-22
> **Responses to Reviewer 69JK**
>
> Thank you very much for your thorough and insightful feedback. We addressed your points here and in the revision. Please let us know if you have further comments.
>
> > I understand that a single model is helpful for multiple UI tasks, but I wonder if this approach is scalable beyond the 5 tasks and 5 modalities mentioned... A discussion on how to add new tasks to the same framework might help, or a discussion on why the current framework is enough.
>
> We believe our framework is sufficient because our input and output modalities are not tied to a specific task, but more related to the generic data type that are often involved in UI modeling tasks. The input includes image and view hierarchy (with focus), and language. The output include view hierarchy, object references, and language responses. Image and view hierarchy define the UI screen, and the focus map directs the model to the specific UI object(s). Language as input is a versatile task descriptor and it as a sequence output can carry various semantics beyond natural languages, e.g., operations for editing a UI. The 5 tasks are representative in UI modeling. It wouldn't be difficult to add new tasks to the model. The model can be trained for general UI-QA tasks, e.g., answering other usability questions than tappability, or for generating a sequence of tokenized edits for revising the UI.
>
> > One of the important motivations of multi-modal multi-task learning mentioned was to achieve better or on-par performance with a single model (and supposedly fewer computations) which is crucial for devices with limited computing resources. But a direct head to head comparison for the computational cost of a multi-task model and individual models is not provided... Such a comparison would highlight the advantages of the multi-task model and would be helpful for the relevant audience.
>
> This is a great point you proposed. We have revised the paper to add the following head-to-head comparison in Appendix E. Overall, the joint model is as fast as the single task model. Although the model size of the joint model is larger than each individual single-task model, the joint model is still much smaller than the total size of all five single models. This shows that VUT substantially reduces the number of models or storage footprint needed for achieving all these task, while obtaining better or on-par accuracy, with little time overhead.
>
> **Inference Time (ms)**
>
> | Task   |      5-Task Model  (ms)    |  Single-Task Model (ms) |
> |----------|-------------|------|
> | Layout | 12.21 | 11.59 |
> | Caption | 30.02 | 30.00 |
> | Summary | 43.68 | 42.97 |
> | Ground | 70.53 | 71.29 |
> | Tap | 51.55 | 51.09 |
>
> **Model Size (millions of params)**
>
> | Task   |      5-Task Model  (M)    |  Single-Task Model (M) |
> |----------|-------------|------|
> | Layout | 63.6 | 39.5 |
> | Caption | 63.6 | 56.2 |
> | Summary | 63.6 | 56.2 |
> | Ground | 63.6 | 63.5 |
> | Tap | 63.6 | 56.2 |
> |5 tasks | 63.6 | 271.6 |
>
> > I also believe that the downstream tasks are also somewhat similar...
>
> We want to clarify that these tasks are very different from each other. UI object detection only takes image input and generates UI objects. Command grounding leverages all the input modalities and outputs an object reference. Tappability, UI summarization and widget captioning use the same model components but the nature of each task is quite different. Tappability is concerned with the usability of a specific UI object. In contrast, UI summarization and widget captioning are to generate functional descriptions about UIs, with one focusing on the entire screen and the other concentrating on a specific object. We have clarified the difference between these tasks in the revision.
>
> > There is no comparison provided with any baseline for 2/5 tasks (Language grounding, Tappability)
>
> These two experiments are based on two new datasets that do not allow a direct comparison with prior work. For Language Grounding, the baseline “Command Grounding Without Image Input” resembles the model previously proposed in Li et al. ACL’20.
>
> > The novelty in the paper is somewhat lacking. The techniques used in the paper ... have been shown to work in the past for image-text tasks [1, 2]. The paper is taking all the lessons from past works and applying it to a new domain.
>
> We agree that our work uses existing components as building blocks to solve a new domain. We have already cited [2] in the original submission and added [1] in the revision. Our work expands the scope of multimodal multi-task learning to the UI domain, while previous work mostly focused on natural images and language. We believe our work is valuable for bringing a new domain and benchmarks to the ICLR community.
>
> > The training procedure mentioned in section 5.2.2 talks about joint training but the procedure followed for training for individual tasks or a subset of tasks is not described in detail...
>
> We clarified the hyperparameter and training schedules in Appendix B.1 in the revision.

---

### Official Review · Reviewer_j2ye · 2021-11-01

**Correctness:** 3
**Technical Novelty And Significance:** 2
**Empirical Novelty And Significance:** 2
**Recommendation:** 5
**Confidence:** 3

**Details Of Ethics Concerns:**

No such concerns.

**Main Review:**

In summary, their main works can be summarized as followed:
(1) The authors formulate multi-modal multi-task learning for graphical user interfaces and design a VUT model.
(2) The authors create a so-called Language Grounding dataset for language grounding task.
This topic is interesting and it is pioneer to incorporate muti-task learning in graphical user interfaces. The manuscript is easy to follow. And the technical contribution is significant. The experimental results conducted on 5 distinct UI tasks demonstrate its superior performance over some state-of-the-art methods.


The reviewer suggests borderline. The weakness of this paper is as follows.
(1)	Tables 5,6,7 and 8 compare the multi-task learning with the single task counterpart. However, their results show that the multi-task learning cannot significantly facilitate the studied tasks. For example, in the Table 5, the setting of “Screen Summarization alone” performs better than the setting “all 5 tasks”. This is contrary to the original intention of this paper. More explanations should be given.
(2)	There are some typos in this manuscript, e.g., “We experiment with VUT on 5 distinct UI task”.


**Summary Of The Paper:**

This paper presents a multi-modal Transformer for multi-task modeling of user interfaces. It is potentially worthy to investigate how to incorporate multiple UI related tasks into a model. The authors hence design a so-called Versatile UI Transformer model which involves three modals inputs to handle five unique tasks.

**Summary Of The Review:**

I am a researcher interested in UI and published many paper of UI on top conferences.

---

> ### Author Response · Authors · 2021-11-22
> **Responses to Reviewer j2ye**
>
> Thank you a lot for giving us constructive feedback, and recognizing that our work "is pioneer to incorporate multi-task learning in graphical user interfaces." We addressed your concerns here and in the revision. Please let us know if you have further questions.
>
> > (1) Tables 5,6,7 and 8 compare the multi-task learning with the single task counterpart. However, their results show that the multi-task learning cannot significantly facilitate the studied tasks. For example, in the Table 5, the setting of “Screen Summarization alone” performs better than the setting “all 5 tasks”. This is contrary to the original intention of this paper. More explanations should be given.
>
> The focus of our work is not to significantly outperform the single task model on accuracy. Instead, we want VUT to perform multiple tasks simultaneously, achieving accuracy on par with the single task model. The advantage of having such a multi-task model is that it reduces the number of models to be developed and the model footprint when deployed. We do see the multi-task model outperforms the single-task baseline on several metrics. For Table 5, “Screen Summarization alone” performs only slightly better on BLEU-1 and BLEU-2, but worse than “All 5 tasks” on all the other metrics. In the revision, we have clarified our motivation in the abstract and introduction, with model size and time cost reported in Appendix E.
>
> In general, multi-task learning is more challenging than single-task learning, and significant improvement on accuracy can be achieved only when tasks have consistent learning objectives. It is an active topic for improving multi-task learning with heterogeneous learning objectives, e.g., “Conflict-Averse Gradient Descent for Multi-task Learning” [Liu et al., NeurIPS 2021] and “Gradient Surgery for Multi-Task Learning” [Yu et al., NeurIPS 2020]. In our work, the 5 tasks that we addressed are quite different from each other. As a result, it is nontrivial that our multi-task model slightly exceeds or performs on par with each single-task model.
>
> > (2) There are some typos in this manuscript, e.g., “We experiment with VUT on 5 distinct UI task”.
>
> We have fixed the typo in the revision.

---

### Official Review · Reviewer_e7Hg · 2021-11-02

**Correctness:** 3
**Technical Novelty And Significance:** 2
**Empirical Novelty And Significance:** 2
**Recommendation:** 5
**Confidence:** 3

**Main Review:**

Strengths
+ Formulated a multi-modal multi-task learning for graphical user interfaces
+ A new two-tower Transformer architecture is designed

Weakness
- By jointly trained on 5 tasks, the model is on par or slight improves over models trained on individual data sets. This does not match the observation on other image-language data joint training where significant improvement is achieved.
- The 5 tasks are closely related, and it is hard to see how the model can generalize to a different task, such as generating/editing new UI layout
- The main architecture of the model is new, while all the building blocks are not. The technical contribution is somewhat lacking.

**Summary Of The Paper:**

A multi-modal multi-task model is proposed for UI understanding. It has an Image-Structure Transformer and Question-Answer Transformer to encode/decode image, structure and language information. The method is evaluated thoroughly on 5 different data sets and achieves good results.

**Summary Of The Review:**

The paper demonstrated an interesting problem addressed by a standard transformer based model. The potential of generalization to more tasks is not clear, and the improvement over single task models falls below expectation.

---

> ### Author Response · Authors · 2021-11-20
> **Responses to Reviewer e7Hg**
>
> Thank you very much for your feedback! We addressed your main points here. Please let us know if you have any further questions.
>
> > By jointly trained on 5 tasks, the model is on par or slight improves over models trained on individual data sets. This does not match the observation on other image-language data joint training where significant improvement is achieved.
>
> Significant improvement can be achieved when tasks and learning objectives are consistent. However, multi-task learning can be challenging when learning objectives or tasks conflict with each other. In previous image-language work as you pointed out, e.g., CLIP (https://openai.com/blog/clip/), the pretrained data (image captions) and downstream data (image class labels converted to captions) are quite relevant, and the multi-task setup does not have conflicting learning objectives. However, it is not the case in our work. The 5 tasks that we addressed are quite different from each other. The layout task is very different from the rest as it only uses the image input. The other four tasks are also significantly different. Even though Widget Captioning and Screen Summarization seem similar, they require the model to focus on different screen information. On top of the challenge posed by these differences, our dataset is much smaller compared to the image-language datasets, thus difficult for the model to learn commonality across tasks. In fact, previous work has dedicated efforts to address the multi-task learning when tasks are inconsistent, e.g., “Conflict-Averse Gradient Descent for Multi-task Learning” [Liu et al., NeurIPS 2021] and “Gradient Surgery for Multi-Task Learning” [Yu et al., NeurIPS 2020].
>
> > The 5 tasks are closely related, and it is hard to see how the model can generalize to a different task, such as generating/editing new UI layout
>
> As discussed for the above question, although our tasks all address UI modeling problems, they are very different in essence. The reviewer brought up a great question about how to generalize to different tasks. We want to point out that our input and output modalities are based on generic data types. The input includes image, view hierarchy, and language. The output heads are equipped with the capability of generating view hierarchy, object references, and language responses. We envision many tasks that are based on these input and output modalities can be potentially learned with our model. For example, the UI layout generation that the reviewer asked can be handled by the Question-Answer model to generate a sequence of tokens for linearized view hierarchies.
>
> > The main architecture of the model is new, while all the building blocks are not. The technical contribution is somewhat lacking.
>
> We agree with the reviewer that the building blocks are not new. Yet, the overall architecture is novel, as the reviewer agreed, and it is valuable for addressing UI modeling problems that are currently under-explored, which expands the scope of multimodal multi-task learning for the ICLR community.

---

> > ### Comment · Reviewer_e7Hg · 2021-11-25
> > **please justify the multi-task learning for inconsistent tasks**
> >
> > As the authors responded, the five tasks considered in this work are not very similar even though they share the same data modalities. Then the question is, if they are not able to benefit from each other, what's the point to train them together? This only makes the model more complex. I am not quite convinced about the goal of this work.

---

> > > ### Author Response · Authors · 2021-11-30
> > > **Justification for multi-task learning**
> > >
> > > Thank you for your follow-up questions! There are two main benefits for training the multi-task model.
> > >
> > > ### Reducing Model Footprints
> > > The advantage of having such a multi-task model is that it reduces the number of models to be developed and the model footprint when deployed for achieving these representative UI modeling tasks. In the revision, we have clarified our motivation in the abstract and introduction, with the model sizes and time costs analyzed in Appendix E.
> > >
> > > ### Improving Accuracy
> > > We consistently observed that involving the Object Detection task in the learning substantially improved the accuracy of each of the four tasks, including Grounding, Widget Captioning, Screen Summarization and Tappability. As shown in Table 4-7, each task + Object Detection outperforms the task alone significantly. This is reasonable because learning for Object Detection allows the model to acquire a good representation of UI screenshots.
> > >
> > > We also see that all-task joint-learning improves the accuracy on several metrics. For example, the all-task model obtains the best accuracy for Widget Caption across all the metrics. That said, we do see that mixing Grounding with language tasks seems to affect the accuracy, which might imply competing learning objectives.
> > >
> > > Please let us know if you have further questions.

---

### Decision · Program_Chairs · 2022-01-20

**Decision:**

Reject

**Comment:**

This paper proposes a neural architecture for tasks involving user interfaces. Tasks involve detecting objects on screen, writing captions about UI components, attribute recognition, etc. The reviewers for this submission found the proposed model to be reasonable and effective. They also found the paper to be well written and easy to understand. However, they did have one major concern, before and after rebuttal: While the model and design choices were reasonable, they questioned if the insights gained from this paper were of interest and relevance to the broader vision community. They also had other concerns/suggestions including adding inference costs and adding more detail, which were addressed in the rebuttal. Another concern was the fact that multi-task did not provide large gains over single task. I agree with the authors in this regard. I think the goal here was to produce a multi-task model that attained at least parity with respect to single task, because a single model would provide large benefits when running on a device, and hence I think this concern was well addressed.

My takeaway from the paper, reviews and discussion, however, continues to focus on the major concern of the reviewers. I think this paper would have benefited from answering at least one (if not more) of the following questions to the reader: (1) Why should the broader community work on this task ? (2) If this task is of limited interest, are there instead, aspects of this task that serve as a useful testbed for multimodal research ? (3) If the task and testbed are not directly applicable, are there new techniques developed in this paper that are broadly applicable to other problems or domains ?

Unfortunately, I think that this paper does not presently address either of these questions strongly to the reader. The paper proposes a method for their task, but readers who aren't directly interested in that end task may find this submission less interesting, in terms of insights for their own work. Given the above, I encourage the authors to address this concern and resubmit. I recommend rejection.